# Translation, Cross-Cultural Adaptation, and Psychometric Properties of the Persian Version of the Measure of Audiologic Rehabilitation Self-Efficacy for Hearing Aids (P-MARS-HA)

**DOI:** 10.3390/audiolres15020031

**Published:** 2025-03-15

**Authors:** Abdolhakim Jorbonyan, Yadollah Abolfathi Momtaz, Mahshid Foroughan, Saeideh Mehrkian

**Affiliations:** 1Department of Geriatric Health, School of Health, Mazandaran University of Medical Sciences, Sari 48157-33971, Iran; a.jourbonian@mazums.ac.ir; 2Malaysian Research Institute on Ageing, University Putra Malaysia, Serdang 43400, Malaysia; 3Iranian Research Center on Aging, The University of Social Welfare and Rehabilitation Sciences, Tehran 19857-13871, Iran; ma.foroughan@uswr.ac.ir; 4Department of Audiology, The University of Social Welfare and Rehabilitation Sciences, Tehran 19857-13871, Iran; sa.mehrkian@uswr.ac.ir

**Keywords:** hearing aids, self-efficacy, cross-cultural adaptation, psychometric properties, elderly

## Abstract

Objectives: This study aimed to translate, cross-culturally adapt, and evaluate the psychometric properties of the Persian version of the Measure of Audiologic Rehabilitation Self-Efficacy for Hearing Aids (MARS-HA) in elderly Iranian adults. Methods and Materials: This cross-sectional study was conducted in Tehran, 2021. Following translation and cross-cultural adaptation, the face validity, content validity, and reliability of the questionnaire were assessed. The Satisfaction with Amplification in Daily Life (SADL) questionnaire was used to assess concurrent validity. Study participants included 300 hearing-aid users aged 60 years and older who completed the research instruments. Data were analyzed using Confirmatory Factor Analysis (CFA), Cronbach’s alpha coefficient, Pearson correlation coefficient, independent *t*-tests, and analysis of variance (ANOVA) in SPSS and AMOS version 24. The significance level was set at *p* ≤ 0.05 for all tests. Results: The mean (SD) age of the participants was 71.38 (8.05) years. The face and content validity of the questionnaire were confirmed by ten experts (CVI > 0.91). The CFA supported the four-factor structure of the questionnaire, and the goodness-of-fit indices indicated that the final model had a good fit. The Cronbach’s alpha for the total questionnaire was 0.93, and for the subscales, it ranged between 0.83 and 0.93. The Pearson’s correlation analysis results showed a positive and significant correlation between MARS-HA and SADL scores (*r* = 0.61, *p* < 0.05), supporting criterion validity. Conclusions: The P-MARS-HA questionnaire demonstrates good validity and reliability and can be used as an appropriate tool for assessing self-efficacy in hearing-aid use among elderly Iranian adults.

## 1. Introduction

Aging is a complex and dynamic process characterized by physical, psychological, and social changes over time. These changes are deeply interconnected, reflecting a continuous process of growth, development, and adaptation until the time of death [1,2]. Many elderly adults experience declines in health and physical function during aging, which can affect their psychological and social well-being and, ultimately, their quality of life [3,4,5]. Age-related hearing loss (ARHL) is one such physical change in aging, typically starting in early adulthood and approximately doubling in prevalence with each decade of life. Epidemiological studies have shown that more than two-thirds of individuals aged 60 and older suffer from Presbycusis. This type of hearing loss results from a combination of physiological changes related to the aging process and other external risk factors, characterized by a gradual loss of sensory hair cells in the inner ear, which are responsible for converting sound into neural signals [6,7]. Research indicates that hearing loss is associated with significant consequences in aging, including dementia, increased risk of falls, reduced physical activity, and other adverse health conditions. These associations may arise through the effects of hearing loss on individuals’ socio-emotional status, such as depression, social isolation, and feelings of loneliness due to communication limitations and barriers [6,8].

Therefore, based on expert recommendations, the use of hearing aids, as the most common approach to compensate for and enhance hearing ability, can significantly control or reduce the negative consequences of presbycusis. Studies have shown that hearing aids not only improve hearing outcomes and health-related quality of life but may also have a protective effect on adverse aging outcomes, including cognitive dysfunction and depressive symptoms [9,10]. However, adopting hearing aids alone is not sufficient; effective and consistent use is essential for success in auditory rehabilitation. Numerous studies have indicated that a significant number of elderly adults do not use their hearing aids regularly or effectively or discontinue their use altogether after some time [11]. One study noted that only about 20% to 50% of elderly adults with hearing impairment use their hearing aids optimally [12].

Studies indicate that adherence to hearing-aid use among older adults is influenced by two categories of factors: (1) technical–auditory factors, including the severity of hearing loss, speech clarity, type of hearing aid, presence of tinnitus, side effects, and environmental noise; and (2) psychosocial factors, such as individual attitudes, motivation, ability to manage hearing aids, and particularly self-efficacy (an individual’s belief in their capacity to address challenges related to hearing aids) [13,14,15,16]. Higher self-efficacy is associated with greater hearing-aid acceptance, satisfaction, and successful use, whereas lower self-efficacy may lead to discontinuation. Self-efficacy is recognized as a critical determinant in sustaining health-related behaviors and improving therapeutic outcomes. Therefore, integrating self-management strategies and enhancing self-efficacy could play a vital role in promoting the successful use of hearing aids among older adults [17,18,19]. To this end, West and Smith (2007), based on Bandura’s self-efficacy questionnaire guidelines [20] and considering potential reasons for non-adherence to hearing-aid use, such as not benefiting adequately from hearing aids in the presence of background noise, difficulty becoming used to using hearing aids, and concerns regarding the mismanagement and incorrect use of hearing aids due to age-related issues in elderly adults, designed the MARS-HA instrument to assess current self-efficacy beliefs related to hearing aids. This tool can help audiologists to better identify factors influencing patients’ success in using hearing aids and, thereby, provide the necessary support to improve treatment outcomes [21].

The MARS-HA instrument includes 24 items and 4 subscales Aided Listening (AL), Basic Handling (BH), Adjustment (Ad), and Advanced Handling (AH) that assess self-efficacy in the mentioned areas and overall self-efficacy in hearing-aid use. For each question, respondents should use a 10-unit scale to indicate how confident they are that they can do that item from 0% (I cannot do this) to 100% (I am sure I can do this); a higher score on this instrument indicates higher self-efficacy. According to West and Smith, scores of 80 percent or higher indicate adequate levels of self-efficacy. The MARS-HA questionnaire has been used in numerous studies, including research by Kelly-Campbell and McMillan (2015), Johnson CE et al. (2018), and Johnson et al. (2024), in which the association of self-efficacy with successful use and satisfaction with hearing aids has been examined [22,23,24]. In addition, the questionnaire has been translated into Spanish [25] and French–Canadian [26], and its psychometric properties have been confirmed. Given the emphasis of previous studies on the point that the MARS-HA instrument can assist in clinical settings during the auditory rehabilitation process, planning training sessions for hearing-aid use, and monitoring the correct and successful use of hearing aids in different time periods, it is necessary to enable its use in other countries as well. However, to date, there is no evidence that a Persian version of this tool is available. Therefore, the objective of the present study is to translate, cross-culturally adapt, and psychometrically evaluate the Persian version of the MARS-HA. This study seeks to create a valid and reliable tool that can be effectively used in assessing hearing-aid self-efficacy among Persian-speaking users and to help improve the quality of life of elderly adults.

## 2. Materials and Methods

### 2.1. Study Design

This cross-sectional study was conducted in 2021 with the aim of cross-cultural adaptation and evaluation of the psychometric properties of the Persian version of the Measure of Audiologic Rehabilitation Self-Efficacy for Hearing Aids (P-MARS-HA).

### 2.2. Participants

In this study, 300 hearing-aid users, aged 60 years or older, residing in Tehran, Iran, were selected as participants. For sampling, the city of Tehran was divided into four areas based on socio-economic development levels: low, low–middle, high–middle, and high. Several audiology clinics were then randomly selected in each area to collect contact information for elderly adults who had received hearing aids from these clinics. Participants were divided into four groups based on their place of residence, and ultimately, 75 individuals from each group were randomly selected (via lottery) as the study sample.

### 2.3. Inclusion Criteria

At least 6 months post-hearing fitting (unilateral and bilateral hearing aids), ability to communicate and sufficient fluency in Persian to answer the questions, voluntary consent to participate in the study, and normal cognitive ability (AMT ≥ 7) [27]. Exclusion Criteria: Unwillingness to continue participation in the study and incomplete questionnaires.

### 2.4. Research Instruments

In this study, data were collected using a demographic information form, the Measure of Audiologic Rehabilitation Self-Efficacy for Hearing Aids (MARS-HA) questionnaire, the Satisfaction with Amplification in Daily Life (SADL) questionnaire (for criterion validity), and the participants’ most recent audiograms.
Measure of Audiologic Rehabilitation Self-Efficacy for Hearing Aids Questionnaire (MARS-HA)West and Smith (2007) designed the Measure of Audiologic Rehabilitation Self-Efficacy for Hearing Aids (MARS-HA) to assess current self-efficacy beliefs related to hearing aids [21]. This scale consists of 24 questions and 4 subscales: 1. Aided Listening (9 items): This subscale assesses an individual’s self-efficacy when listening with a hearing aid in a specific listening situation; for example, “I could understand a one-on-one conversation in a quiet place if I wore hearing aids”. 2. Basic Handling (7 items): This includes questions assessing an individual’s self-efficacy in the basic management of a hearing aid, for example, “I can insert a battery into a hearing aid with ease”. 3. Adjustment (3 items): Questions assessing an individual’s self-efficacy in adjusting to the hearing aid, for example, “I could get used to the sound quality of hearing aids”. 4. Advanced Handling (5 items): Questions assessing a hearing-aid user’s self-efficacy in performing advanced hearing aid tasks, for example, “I can troubleshoot a hearing aid when it stops working”. For scoring the questions, respondents should report how much confidence they have in their ability on a 10-unit interval scale (0, 10, 20...100 percent) for each question. If they feel they do not have the ability or skill, they score zero, and if they are completely confident in performing that skill, they consider a score of 100 for themselves. A higher score on this questionnaire indicates greater self-efficacy. According to West and Smith, if the average score of the questionnaire is higher than 80 percent, it indicates a sufficient level of self-efficacy. West and Smith reported the reliability of this instrument for experienced hearing-aid users, using Cronbach’s alpha coefficient for the total scale as 0.91, and for the subscales “Basic Handling”, “Advanced Handling”, “Adjustments”, and “Aided Listening” as 0.88, 0.67, 0.73, and 0.91, respectively. In addition, using the test–retest method, the reliability of the instrument was obtained as 0.88 [21].Satisfaction with Amplification in Daily Life (SADL)The Satisfaction with Amplification in Daily Life questionnaire is a valid instrument designed by Cox and Alexander (1999) [28]. This instrument assesses an individual’s satisfaction in various aspects of hearing-aid use and consists of 15 items and four subscales: Positive Effects, Services and Costs, Negative Features, and Personal Image. The average score obtained from each subscale assesses the individual’s range of satisfaction and forms the overall questionnaire score. The method of responding to questions is based on a 7-option Likert scale (from completely agree to completely disagree), such that in four questions (2, 4, 7, 13), scoring is reversed. In the research by Faraji Khiavi et al. (2014), the face and content validity of the Persian version of this tool were reviewed and confirmed by five experts, and its internal reliability was obtained as 0.8 through the calculation of Cronbach’s alpha index [29].

### 2.5. Description of the Procedure

Phase (1): Translation and Localization

In the translation and localization process, initial approval was obtained from the original designers of the questionnaire by sending an email explaining the purpose of the psychometric evaluation and the preparation of the Persian version of the instrument. In the next step, using the forward–backward translation method, the original (English) version of the questionnaire was translated into Persian by two Persian-speaking translators, one of whom was proficient in the terminology and translation of audiology texts. Subsequently, the translations were compared, and the questions were reconciled in terms of meaning and concept. By selecting the best options, a Persian version of this instrument was ultimately prepared. To ensure the complete alignment of the Persian translation with the original text and the fluency of the sentences, the final version of the questionnaire was back-translated into the original language by another translator who was fluent in English and had not previously seen the original questionnaire. At this stage, the English versions were compared, and necessary corrections were applied to the Persian version of the questionnaire under the supervision of translators and experts (including five audiologists and five gerontologists, all holding Ph.D. degrees). Finally, the final Persian version of the questionnaire was obtained.

Phase (2): Psychometric Properties

After translating and preparing the Persian version of The Measure of Audiologic Rehabilitation Self-Efficacy for Hearing Aids Questionnaire (P-MARS-HA), the psychometric properties of the questionnaire, including face and content validity, floor and ceiling effects, reliability (internal consistency and stability), construct validity (confirmatory factor analysis), and criterion validity, were evaluated and examined.

### 2.6. Data Analysis

In this study, descriptive statistics, including frequency distribution (percentage), mean, and standard deviation, were used to describe the demographic characteristics of the study sample. For inferential data analysis, considering the normal distribution of data based on skewness and kurtosis tests (in the range of −2 to +2), independent *t*-tests (to compare the means of two groups), one-way analysis of variance (ANOVA) (to examine differences between means in multiple groups), and Pearson correlation coefficient (to examine the relationship between variables) were used. Data analysis was performed using SPSS (version 24) and AMOS (version 24) software, and the statistical significance level was set at *p* < 0.05 for all tests.

## 3. Results

### 3.1. Floor and Ceiling Effects

The results indicated that a small percentage of participants were at the extreme ends of the score range; specifically, only 2% achieved the lowest score, and 2% achieved the highest score. This balanced distribution of scores suggests that the questionnaire used did not have significant floor or ceiling effects.

### 3.2. Descriptive Results

In this study, 300 elderly hearing-aid users participated. The mean (SD) age of the participants was 71.38 (±8.05) years. Regarding gender, half of the participants were female (50.7%). Most participants were married (70.3%), and 81% of them used digital hearing aids (Table 1). The mean (SD) overall P-MARS-HA scale score was 65.24 (±15.94). The lowest mean score was for the Advanced Handling (AH) subscale, and the highest mean score was for the Basic Handling (BH) subscale (Table 2). The correlation of all subscales with the overall questionnaire mean was positive and significant using Pearson’s correlation coefficient (*p* < 0.05) (Table 2). The correlation between the P-MARS-HA scale and participants’ age was negative and significant (*r* = −0.26, *p* < 0.05), meaning that as age increases, the self-efficacy of elderly adults in using hearing aids decreases. Independent *t*-test results showed that In-The-Ear (ITE) hearing-aid users had significantly higher self-efficacy compared to Behind-The-Ear (BTE) hearing-aid users (*p* < 0.05), but this difference was not significant for gender and type of hearing-aid prescription (unilateral or bilateral) (*p* > 0.05).

Analysis of variance results showed that the mean P-MARS-HA questionnaire score differed significantly across groups based on education, marital status, and degree of hearing loss in the worse ear (WEA), and type of hearing aid (*p* < 0.05) (Table 3). The Scheffe post hoc test results showed that mean self-efficacy was significantly higher in married individuals compared to single individuals, in individuals with a university education level compared to illiterate and elementary-educated individuals, in individuals with mild to moderate hearing loss compared to severe to profound hearing loss, and in digital hearing-aid users compared to analog and programmable hearing-aid users (*p* < 0.05).

### 3.3. Face Validity

To assess the face validity of the P-MARS-HA questionnaire, 10 experts in the fields of gerontology and audiology were asked to evaluate the clarity, comprehensibility, and relevance of each question to the main objective of the questionnaire. The results indicated that all questions, from the perspective of these experts, had sufficient clarity and relevance and adequately covered the objective of the questionnaire. Additionally, the impact score of all questions was reported to be higher than 1.5, which is another confirmation of the favorable face validity of the questionnaire. Based on these findings, no need was identified to remove or modify the questionnaire questions at this stage.

### 3.4. Content Validity

The content validity of the questionnaire was assessed by evaluating ten experts in the fields of gerontology and audiology. Experts evaluated each of the questionnaire’s items based on the criteria of necessity and importance, and then the content validity ratio (CVR) and content validity index (CVI) were calculated. The CVR and CVI values for all items were higher than 0.80 and 0.91, respectively, indicating that the P-MARS-HA instrument has acceptable content validity.

### 3.5. Construct Validity (Confirmatory Factor Analysis)

The construct validity of the questionnaire was assessed using a confirmatory factor analysis (CFA). After performing modifications, the final model had an acceptable fit. Factor loadings are within the acceptable range (>0.5) (Table 4), and the final model is presented in the diagram (Figure 1). In this study, the goodness-of-fit index of the chi-squared per the number of degrees of freedom (χ^2^/df) was 4.02, indicating an acceptable model fit. The values of other fit indices are also reported in Table 5. Another noteworthy point is that the correlation between the items of the instrument was significant and positive (*p* < 0.05).

### 3.6. Criterion Validity

To assess criterion validity, the concurrent validity method was used, whereby the correlation between the MARS-HA questionnaire and Satisfaction with Amplification in Daily Life (SADL) was evaluated. Pearson’s correlation analysis results showed that there is a positive and significant correlation between MARS-HA and SADL scores (*r* = 0.61, *p* < 0.05). This finding indicates that individuals with higher self-efficacy in using hearing aids have greater satisfaction with their hearing aids. Given that the correlation of 0.61 is in the medium to strong range, it can be concluded that the P-MARS-HA questionnaire has predictive criterion validity and can be used as a tool to predict the level of satisfaction of individuals with hearing aids.

### 3.7. Reliability

The internal reliability of the P-MARS-HA questionnaire was assessed by calculating the Cronbach’s alpha coefficient. The Cronbach’s alpha value for the entire questionnaire was 0.93, indicating good reliability. The alpha values for the subscales were slightly different: Basic Handling: 0.90, Advanced Handling: 0.83, Adjustment: 0.87, and Aided Listening: 0.93 (Table 2). The analysis of the correlation of each question with the rest of the questions also showed that all questions correlate higher than 0.4 with each other. Therefore, the reliability of this questionnaire seems desirable.

## 4. Discussion

Assessing self-efficacy in hearing-aid use can help identify psychological and practical barriers to hearing-aid use. Additionally, it can provide strategies to increase the acceptance and effective use of hearing aids in elderly adults. Therefore, the present study aimed to translate, cross-culturally adapt, and examine the psychometric properties of the Persian version of The MARS-HA. A total of 300 hearing-aid users aged 60 years and older participated in this study. The results of this study indicated that the Persian version of the MARS-HA scale has a favorable translation and acceptable psychometric properties in elderly adult Persian-speaking hearing-aid users.

In the descriptive–analytical findings of the study, the mean (SD) of the Persian version of the MARS-HA was 65.24 (±15.94), and there was a positive and significant relationship between all subscales. Consistent with this finding, the original version and the Spanish and French–Canadian versions also showed a positive and significant relationship between the subscales. The obtained mean was equal to the mean of the original questionnaire and lower than the mean of other versions of the questionnaire [21,25,26]. Among the subscales, the lowest mean was related to Advanced Handling, and the highest mean was related to Basic Handling. Consistent with this finding, other versions of this questionnaire also showed that Advanced Handling had the lowest mean [21,25,26]. It seems that the self-efficacy of elderly adults in performing tasks that require more complex knowledge, greater cognitive ability, and finer motor skills, such as troubleshooting, cleaning, and caring for the hearing aid, is lower than that for simpler and more repetitive tasks, such as inserting the battery into the hearing aid and placing the hearing aid on the ear. In this regard, strengthening self-efficacy through technical training, support services, and encouraging elderly adults to practice and perform these tasks can be effective.

The findings of the study showed that individuals with older age, individuals with lower education, individuals who are single, and individuals with a greater degree of hearing loss had lower self-efficacy. Consistent with our study, in the study by Fuentes et al. (2019), individuals with higher education and a lower PTA (Pure tone average) had greater self-efficacy in hearing-aid use. However, in that study, age was not significantly associated with self-efficacy in hearing-aid use and was contrary to the findings of our study [25]. Individuals with higher education usually have high activity and social participation, and their access to information is greater. In addition, they have a greater ability to understand the technical instructions of hearing aids. These factors can increase self-efficacy in individuals with higher education. In the study by Johnson & Sarangi (2023), it was reported that factors such as family encouragement and social support can have a significant impact on self-efficacy in hearing-aid use [22]. In our study, it was also found that married individuals, compared to single individuals who are usually deprived of receiving direct support from their spouse, have greater self-efficacy in hearing-aid use. Studies have shown that with increasing age, elderly adults, due to a decline in cognitive abilities (memory, attention) and physical abilities (fine motor skills), may have difficulty performing complex tasks [30,31]. Therefore, their self-efficacy in performing more complex hearing-aid tasks may be lower. On the other hand, concern about damaging the hearing aid and lack of access to appropriate training can also exacerbate this issue. In terms of the degree of hearing loss, it seems that individuals with more severe hearing loss, due to the functional limitations of hearing aids in noisy environments, technical challenges and more complex settings of more powerful hearing aids, and the lower participation of individuals with severe hearing loss in activities, have less self-efficacy in using hearing aids.

The validity and reliability of the Persian version of MARS-HA in this study were approximately like the original version and other versions of this questionnaire. In terms of reliability, the Cronbach’s alpha coefficient obtained from the Persian version of the MARS-HA questionnaire was 0.93, and for the subscales, it was higher than 0.8, which was similar to the original version and slightly higher than the Spanish and French–Canadian versions [21,25,26].

The results of the present study provide strong evidence in support of the construct validity of the Persian version of the MARS-HA questionnaire. Confirmatory factor analysis (CFA) showed that the factor structure of the original version was also preserved in the Persian sample. The values of the fit indices also indicate that the constructs of the questionnaire are understood and experienced in the cultural context of Iran in the same way as they are defined in the original culture of the questionnaire. Factor loadings higher than 0.5 and significant for all items in the relevant factor indicate that items well measure their intended construct. Also, the significant and expected correlations between the different factors of the questionnaire confirm the convergent validity of this instrument. Regarding predictive criterion validity, a positive and significant relationship was obtained in this study between P-MARS-HA and SADL. In confirmation of this finding, in the studies of Kelly et al. (2015) and Ferguson et al. (2016), MARS-HA had a positive and significant relationship with SADL [18,24]. It seems that people who have high self-efficacy in using hearing aids are better able to cope with challenges, such as adjusting the device, adapting to noisy environments, or communicating in groups. This confidence increases the feeling of control over life and enhances satisfaction with the hearing aid. In summary, the MARS-HA questionnaire is recognized as an instrument with high potential in clinical and research settings. This questionnaire can be useful in guiding the auditory rehabilitation process, planning initial hearing-aid familiarization sessions, and determining the goals of subsequent visits for patients using hearing aids [21]. The use of P-MARS-HA as a clinical tool can lead to better outcomes in hearing-aid use, including increased self-efficacy in hearing-aid use. Finally, successful use of hearing aids can lead to an improved quality of life for elderly adults.

## 5. Conclusions

According to the results of the study, the Persian version of the MARS-HA scale has acceptable validity and reliability in the Iranian community, and it can be used to assess self-efficacy in hearing-aid use in elderly adults using hearing aids. This instrument, with its four subscales, covers various dimensions of self-efficacy related to hearing-aid use, which include Advanced Handling, Basic Handling, Aided Listening, and Adjustment. The use of this tool enables hearing specialists to gain a better understanding of level of self-efficacy of elderly adults in using hearing aids, and based on that, design and implement more appropriate auditory rehabilitation programs.

## 6. Clinical Applications

The P-MARS-HA facilitates optimized hearing-aid outcomes by assessing self-efficacy across four domains: Basic Handling, Adjustment, Aided Listening, and Advanced Handling. Administration of the tool is recommended 2–4 weeks post-fitting to identify challenges during initial acclimatization, with follow-up assessments suggested at 3- and 6-month intervals. Low self-efficacy scores guide targeted interventions, such as hands-on training, scenario-based counseling, communication strategy practice, and psychosocial support. Integrating MARS-HA into clinical workflows—through pre-visit screenings, SMART goal-setting, and multidisciplinary care—establishes a personalized, data-driven approach. Clinical studies demonstrate that this integration enhances hearing-aid adherence, reduces device abandonment, and improves patients’ quality of life by addressing practical and psychosocial barriers to successful rehabilitation.

## 7. Limitations

This study has some limitations that should be considered. First, the cross-sectional design of the study does not allow for examining changes in self-efficacy over time and their impact on related outcomes. Second, while other versions of the questionnaire have compared the self-efficacy of experienced and novice hearing-aid users, the present study only examined users who had been using hearing aids for at least six months. This may limit the generalizability of some of the results to all hearing-aid users, particularly in the early stages of hearing-aid use.

## Figures and Tables

**Figure 1 audiolres-15-00031-f001:**
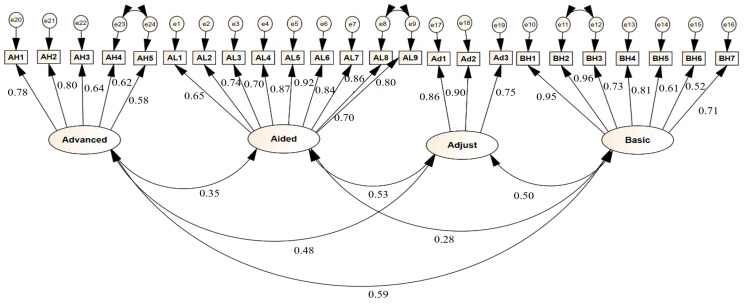
The final measurement model of the research and its parameters using standardized coefficients. AH: Advanced Handling, AL: Aided Listening, Ad: Adjustment, BH: Basic Handling.

**Table 1 audiolres-15-00031-t001:** Demographic characteristics of the participants (N = 300).

Variables		Frequency (%)
Age	Mean (SD):	71.38 (8.05)
Gender	Female	152 (50.7%)
	Male	148 (49.3%)
Education status	Illiterate	45 (15%)
	Primary	66 (22%)
	Secondary	65 (21.7%)
	Diploma	77 (25.7%)
	Higher education	47 (15.6%)
Marital Status	Married	211 (70.3%)
	Widow	71 (23.7%)
	Divorced	18 (6%)
Types of Hearing Aids	Digital	243 (81%)
	Programmable	37 (12.3%)
	Analog	20 (6.7%)
Degree of hearing impairment (WEA)	˂26 dB HL	0
	26–40 dB HL	9 (3%)
	41–55 dB HL	82 (27.3%)
	56–70 dB HL	149 (49.7%)
	71 dB HL	60 (20%)
Hearing aid fitting	Unilateral	98 (32.7%)
	Bilateral	202 (67.3%)
Style of hearing aid	Behind the ear (BTE)	203 (67.7%)
	In-the-ear (ITE)	97 (32.3%)

WEA = worse ear average (averaged over 0.5, 1, 2, and 4 kHz).

**Table 2 audiolres-15-00031-t002:** Correlation coefficients and reliability of P-MARS-HA scores (*n* = 300).

Variables	Mean (SD)	Basic Handling	Advanced Handling	Adjustment	Aided Listening	Overall Score
Basic handling	73.76 (21.94)	1				
Advanced handling	40.03 (23.05)	0.67 *	1			
Adjustment	67.01 (24.97)	0.46 *	0.45 *	1		
Aided listening	72.03 (17.39)	0.29 *	0.30 *	0.48 *	1	
Overall score	65.24 (15.94)	0.81 *	0.78 *	0.71 *	0.71 *	1
Reliability (Cronbach’s Alpha)	-	0.90	0.83	0.87	0.93	0.93

*p* ˂ 0.05 *.

**Table 3 audiolres-15-00031-t003:** Statistical comparison of sociodemographic subgroups (*n* = 300).

Variables.		Mean (SD)	T/F Statistic	*p*-Value
Gender	Male	64.77 (14.45)	−0.49	0.62
	Female	65.69 (17.30)		
Marital status	Single	60.40 (17.05)	4.75	0.009 *
	Married	66.48 (15.21)		
	Divorced or widowed	69.76 (16.54)		
Education	Illiterate	60.37 (18.25)	5.36	0.001 *
	Primary	60.19 (14.24)		
	Secondary	67.05 (15.78)		
	Diploma	66.94 (15.25)		
	Higher education	71.70 (14.15)		
Degree of hearing impairment (WEA)	26–40 dB HL	66.88 (21.56)	4.91	0.002 *
	41–55 dB HL	67.44 (13.34)		
	56–70 dB HL	66.74 (16.03)		
	71 dB HL	58.34 (16.63)		
Hearing-aid fitting	Unilateral	66.17 (16.92)	0.7	0.48
	Bilateral	64.79 (15.42)		
Types of hearing aids	Digital	66.61 (15.21)	4.91	0.008 *
	Programmable	60.13 (18.54)		
	Analog	58.12 (17.50)		
Style of hearing aid	Behind the ear (BTE)	62.14 (16.30)	25.74	0.001 *
	In-the-ear (ITE)	71.73 (12.99)		

*p* < 0.05 *.

**Table 4 audiolres-15-00031-t004:** Goodness-of-fit indices for the measured model of the P-MARS-HA.

Model	χ^2^/df	CFI	PCFI	RMSEA	GFI	IFI
Recommended value	˂5	≥0.9	˃0.5	<0.1	≥0.9	0–1
P-MARS-HA	4.02	0.93	0.67	0.09	0.901	0.89

χ^2^/df: Chi-square test; CFI: comparative fit index; PCFI: parsimony comparative fit index; RMSEA: root mean square error of approximation; IFI: incremental fit index; GFI: goodness-of-fit index.

**Table 5 audiolres-15-00031-t005:** Regression coefficients of the paths.

Path		Standardized Regression Coefficient	Standard Error	C.R Statistic	*p* Value
1	AL1<— Aided listening	0.64			
2	AL2<— Aided listening	0.74	0.10	11.36	0.001 *
3	AL3<— Aided listening	0.69	0.12	10.76	0.001 *
4	AL4<— Aided listening	0.87	0.13	12.90	0.001 *
5	AL 5<— Aided listening	0.92	0.14	13.45	0.001 *
6	AL 6<— Aided listening	0.84	0.13	12.58	0.001 *
7	AL 7<— Aided listening	0.85	0.13	12.73	0.001 *
8	AL8<— Aided listening	0.79	0.13	12.09	0.001 *
9	AL 9<— Aided listening	0.69	0.12	10.74	0.001 *
10	BH1<— Basic handling	0.94			0.001 *
11	BH2<— Basic handling	0.95	0.02	34.38	0.001 *
12	BH3<— Basic handling	0.72	0.05	16.32	0.001 *
13	BH4<— Basic handling	0.80	0.03	20.68	0.001 *
14	BH5<— Basic handling	0.61	0.04	12.61	0.001 *
15	BH6<— Basic handling	0.51	0.05	10.03	0.001 *
16	BH7<— Basic handling	0.71	0.04	16.05	0.001 *
17	Ad1<— Adjustment	0.86			0.001 *
18	Ad2<— Adjustment	0.90	0.06	18.40	0.001 *
19	Ad3<— Adjustment	0.74	0.05	14.74	0.001 *
20	AH1<— Advanced handling	0.77			0.001 *
21	AH2<— Advanced handling	0.79	0.07	12.76	0.001 *
22	AH3<— Advanced handling	0.63	0.06	10.38	0.001 *
23	AH4<— Advanced handling	0.62	0.08	10.31	0.001 *
24	AH5<— Advanced handling	0.58	0.05	9.37	0.001 *

AH: Advanced Handling, AL: Aided Listening, Ad: Adjustment, BH: Basic Handling, *p* ˂ 0.05 *.

## Data Availability

Since this paper is extracted from a thesis, open access to the raw data is not permitted by the Medical University. If necessary, the authors can provide access to the data with the permission of the university.

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
