# Peer review of "Translation, Cross-Cultural Adaptation, and Psychometric Properties of the Persian Version of the Measure of Audiologic Rehabilitation Self-Efficacy for Hearing Aids (P-MARS-HA)"

_audiolres, 2025, doi:10.3390/audiolres15020031_

Round 1
Reviewer 1 Report
Comments and Suggestions for Authors
Thank you so much for the possibility to read and review your work.
This work deserves to be known by the scientific and clinical population in the Persian language, once represents a very good methodological research design and lacks of big mistakes or lacks of information.
Because of that, I propose to do some improvements to your paper. Please see de document I attached with my revision notes. Thank you

Some errors that I signalled in the document attached, for example Aging instead of ageing....
Author Response
Comment 1: [Suggestions for Lexical Refinements]
Response 1: [We appreciate your suggestions regarding the refinement of certain terms, such as "ageing." We have incorporated all the revised terms you recommended into our text, ensuring that our manuscript is both precise and fluent. Thank you for your valuable feedback, which has significantly enhanced the clarity and readability of our work. We have highlighted the revised terms in red].
comment2: [Why? If the questionnaire is not complete should be excluded]
Response 2: [We appreciate your meticulous attention to detail and valuable feedback.
In response to your question, in fact, there were no incomplete questionnaires in our final dataset. Nevertheless, to ensure the quality of the data, our exclusion criteria included questionnaires with more than 5% of unanswered questions. Given that no questionnaire actually violated this criterion, this aspect of the exclusion criteria was not implemented in practice. Therefore, we will remove this statement from the exclusion criteria section to enhance the clarity of the text. (line-126)]
Comment 3: [I can't find the characterisation of the expert panel with the criteria of inclusion...]
Response 3: [ten distinguished experts. This included five audiologists and five gerontologists, all of whom held Ph.D. degrees and were faculty members at medical universities. These experts assisted us in the translation and face validity of the questionnaire. Their participation enhanced the quality of our study. We have added the characteristics of the expert panel to the text and highlighted them in red (line-186)]
Comment 4: [Why the author is not the person in charge of evaluating the back translation to English?]
Response 4:[ As described in our methodology, the back translation was performed by a translator who was fluent in English and had not previously seen the original questionnaire. This ensured that the evaluation was unbiased and focused solely on ensuring that the Persian translation accurately reflected the original English version. Specifically, the authors participated in the reconciliation and finalization of the Persian version of the questionnaire to ensure that it met the required standards.]

Reviewer 2 Report
Comments and Suggestions for Authors
It is an interesting article. I do not have any specific recommendations, which could be mistaken as negative. I will present my commends step by step
1, Introduction. This part is very long. I believe that paragraphs 3 and 4 could be rewritten and transferred in the Discussion.
2, Methods and purposes are sufficiently presented.
3. Results. Some diagrams could be helpful and improve the whole text.
4. The discussion should be re-written, taking into consideration paragraphs written in the Introduction.
5. The References are o.k
Author Response
Comment 1: [Introduction. This part is very long. I believe that paragraphs 3 and 4 could be rewritten and transferred in the Discussion. 4. The discussion should be re-written, taking into consideration paragraphs written in the Introduction.]
Response 1: [Thank you for your insightful feedback. We agree with your observation that the Introduction was overly lengthy. To address this, we condensed the content of paragraphs 3 and 4, which elaborated on technical-auditory and psychosocial factors influencing hearing aid adherence, and incorporated a detailed explanation of these concepts into a new subsection titled “Clinical Applications” at the end of the manuscript. The revised and added sections have been highlighted in green for ease of identification. This restructuring ensures alignment between the theoretical framework and the practical interpretations of our findings while avoiding redundancy. We appreciate your constructive critique, which has significantly enhanced the clarity and focus of the manuscript. (Line: 68& 384)]
Comment 2: [Results. Some diagrams could be helpful and improve the whole text.]
Response 2: [Thank you for your valuable suggestion. We acknowledge that incorporating diagrams could enhance the interpretability of the results. However, our initial strategy aimed to maintain conciseness in the manuscript by limiting the number of tables and figures to avoid excessive length. That said, if you deem the addition of diagrams essential for clarity, we are happy to include them in the revised version. We appreciate your understanding and would like to emphasize that in our upcoming studies, we plan to conduct a more in-depth analysis of self-efficacy in hearing aid adherence, employing advanced statistical methods to further elucidate this critical concept. Your feedback is instrumental in guiding our future research directions.]

Reviewer 3 Report
Comments and Suggestions for Authors
Dear Authors, thank you for your manuscript. I have attached some comments in the PDF for your consideration. The validation and construct of this translated version of the MARS-HA is sound and rigorous. Although not very novel as this is an adaptation of existing valid questionnaire of hearing aid self-efficacy, this still remains an important clinical tool for monitoring hearing aid rehabilitation outcomes. Perhaps authors can expand a little more on how the tool can be used to inform clinicians on developing targeted strategies to improve hearing aid self-efficacy and also review the literature to see if there are existing studies using MARS-HA to inform hearing aid rehabilitation strategies. This will help to strengthen the significance of culturally adapting this questionnaire as the final goal should be to improve hearing aid outcomes!

Author Response
Comment 1: [sentence appears to be incomplete. Do you mean and adaptation (without the word until)]
Response 1: [Thank you for your insightful comment. We appreciate your suggestion regarding the sentence structure. The phrase "until the time of death" was likely omitted due to an error, and we have now reinstated it to provide clarity. We have highlighted the revised section in yellow for your reference.(line-42).]
Comment 2: [Will be good to expand on what constitutes the different SES status- by financial household income I suppose but should have some details]
Response 2: [We appreciate your attention to detail. The division of Tehran's socio-economic development levels into different regions was based on a study that utilized 16 socio-economic indicators. These indicators included housing quality, urban welfare, a composite development index, and an economic index, among others. Our study used this division for sampling purposes.]
Comment 3: [Curious if there were any post-hoc sensitivity analyses performed to look into the older age group with lower self-efficacy. was it generally lower education levels or higher, what were the other parameters in the older age group that could explain poorer self-efficacy besides age since age alone is probably not the best predictor. In terms of the various factors in explaining the variance of self-efficacy as on outcome measure, which factors had higher weightage in contributing to self-efficacy? Sugguestion for some further clarifications here.]
Response 3: [Thank you for your thoughtful comment and for highlighting the importance of self-efficacy in hearing aid use. In this study, due to the psychometric nature of the questionnaire, we placed less emphasis on demographic factors and their relationship with self-efficacy. However, we are currently preparing a separate manuscript focused on the factors associated with hearing aid self-efficacy. This upcoming study will provide a more detailed analysis of the relationships between demographic variables (e.g., education level), as well as other factors such as hours of hearing aid use, self-reported hearing disability, satisfaction with hearing aids, and additional parameters. The results of this research will address the questions and uncertainties you have raised comprehensively.]
Comment 4: [You have made these inferences in general, how about specific in context to your own participants? Was there any data logging of the hearing aids to show hours of use or environmental acoustical data collected from hearing aids to show that mostly participatns were hearing in noise in those non-users/poorer users with lower self-efficacy?]
Response 4: [This study formed part of a thesis project in which we systematically examined factors associated with regular hearing aid use among older adults with hearing loss. Our findings revealed a significant association between lower self-efficacy levels and reduced hearing aid adherence. However, as the primary objective of this study was to evaluate the psychometric properties of a self-efficacy questionnaire, we deliberately omitted detailed analyses of hearing aid usage patterns (e.g., logged usage hours or environmental acoustical data). Future studies may benefit from incorporating such objective metrics to further elucidate the interplay between self-efficacy, environmental factors, and hearing aid utilization.]
Comment 5: [How do the authors suggest we use P-MARS-HA or MARS-HA in general as a tool to improve outcomes in hearing aid rehabilitaiton? When (how soon) should it be administered after hearing aid fitting and at follow up if MARS-HA self efficacy is poor, how should clinicians address the problem? A slightly expanded segment on using this clinical tool to suggest targeted improvements in hearing rehabilitaiton will be good]
Response 5: {Thank you for your insightful suggestion. We found your recommendation valuable and have added a dedicated section titled "Clinical Applications" to the manuscript. This section includes insights that can be beneficial for audiologists, caregivers, and hearing aid users. We believe this addition strengthens the translational impact of our findings, with the added section highlighted in yellow.(line-385)
Clinical Applications
The P- MARS-HA facilitates optimized hearing aid outcomes by assessing self-efficacy across four domains: Basic Handling, Adjustment, Aided Listening, and Advanced Handling. Administration of the tool is recommended 2–4 weeks post-fitting to identify challenges during initial acclimatization, with follow-up assessments suggested at 3- and 6-month intervals. Low self-efficacy scores guide targeted interventions, such as hands-on training, scenario-based counseling, communication strategy practice, and psychosocial support. Integrating MARS-HA into clinical workflows—through pre-visit screenings, SMART goal-setting, and multidisciplinary care—establishes a personalized, data-driven approach. Clinical studies demonstrate that this integration enhances hearing aid adherence, reduces device abandonment, and improves patients’ quality of life by addressing practical and psychosocial barriers to successful rehabilitation.]
